# Transcriptomic Analysis of Liver in Silver sillago, *Sillago sihama* Fed with High-Level Low-Gossypol Cottonseed Meal in Replacement of Fishmeal Diet

**DOI:** 10.3390/ani13071194

**Published:** 2023-03-29

**Authors:** Hao Liu, Menglong Zhou, Xiaohui Dong, Beiping Tan, Shuang Zhang, Yuanzhi Yang, Shuyan Chi, Hongyu Liu, Xiaobo Yan, Zhihao Li

**Affiliations:** 1Laboratory of Aquatic Animal Nutrition and Feed, College of Fisheries, Guangdong Ocean University, Zhanjiang 524088, China; 2Aquatic Animals Precision Nutrition and High-Efficiency Feed Engineering Research Centre of Guangdong Province, Zhanjiang 524088, China

**Keywords:** transcriptomic, *Sillago sihama*, low-gossypol cottonseed meal, liver, fishmeal

## Abstract

**Simple Summary:**

Understanding the molecular mechanisms involved in adapting to alternate diets is important due to the increasing substitution of fish meal protein with plant protein in aquafeeds. To assess the effects of a diet with 64% low-gossypol cottonseed meal on juvenile *Sillago sihama* (*S. sihama*), growth performance, quality, liver function, and RNA-sequencing (RNA-seq) were evaluated and compared to those fed a traditional FM-based diet. Results showed that indicators of growth were lower in the 64% LCSM (R64) group, with significant differences in whole crude lipid and liver histology. Liver transcriptome analysis indicated that high LCSM intake affected lipid and amino acid metabolic pathways, as well as hepatic gluconeogenesis and glycolysis. This study highlights the harmful effects of feeding high levels of LCSM to *S. sihama*, and suggests that it should not be used as a substitute for high levels of FM in their diet.

**Abstract:**

Understanding the molecular mechanisms involved in adaptation to alternate diets has become a significant concern, as increasing amounts of fishmeal (FM) protein in aquafeeds are being substituted with plant protein. Thus, the goal of this study was to assess growth performance, quality, and liver function of juvenile *Sillago sihama* (*S. sihama*) through growth indices, whole-body composition, histology of the liver, and RNA-sequencing (RNA-seq), after they were fed a formulated diet with 64% low-gossypol cottonseed meal (LCSM) for 56 days, compared to those fed a traditional FM-based diet. Indicators of growth, including final body weight (FBW), weight gain rate (WGR), specific growth rate (SGR), protein efficiency ratio (PER), and condition factor (CF), were considerably lower in the 64% LCSM (R64) group than in the FM diet group. In the R64 diet, the whole crude lipid was significantly lower than in the FM diet. The hematoxylin–eosin section showed that dietary high levels of LCSM resulted in diffuse lipid vacuolation in the liver of *S. sihama*. According to a liver transcriptome analysis, high LCSM intake in the diet significantly impacted lipid synthesis and catabolism, elevated pathways for cholesterol synthesis, blocked several amino acid metabolic pathways, and adversely affected hepatic gluconeogenesis and glycolysis. The findings of this study indicate that feeding high levels of LCSM in *S. sihama* is harmful to the growth of the organism and can harm the liver’s structural integrity, as well as obstruct the normal metabolism of amino acids, lipids, and carbohydrates. Therefore, it is not recommended to substitute LCSM for high levels of FM in the diet of *S. sihama*.

## 1. Introduction

Fishmeal (FM) is increasingly in demand due to its optimal nutritional qualities for fish diets, as a result of the rapid rise of worldwide aquaculture production [1]. However, fishmeal resources are limited and prices are rising steadily. Finding alternative sources of protein has become a necessity for sustainable aquaculture [2]. Plant protein (PP) has been used as an alternative protein source due to its relatively abundant availability and low price.

Although all-PP diets have been successfully applied to some omnivorous and herbivorous fish [3,4,5], and can even be accepted by several carnivorous species [6,7,8,9], most carnivorous species are more sensitive to the negative aspects of PP, including poor palatability, essential nutrient deficiencies, and antinutritional factors [10,11].

With a body length of up to 30 cm, the silver sillago (*Sillago sihama* Forskál, *S. sihama*) belongs to the Perciformes, Sillago family, and *Sillago* genus, and is mostly found in the coastal waters of the Indian Ocean and the eastern coast of the Pacific Ocean [12]. It is highly loved for its outstanding flesh quality, high economic and nutritional value, and plays a major part in fishing activities in China’s coastal waters as an important economic fish in China’s coastal districts [13]. However, due to overfishing, the natural resources of *S. sihama* have been dropping in recent years, and market prices have increased, resulting in younger individuals, smaller groups, and decreased yields [14]. Since 2007, research on *S. sihama* artificial breeding has been conducted, and in 2012, Guangdong Ocean University effectively broke through the key technology of *S. sihama* artificial breeding [15]. At the moment, studies on *S. sihama* are mostly focused on morphology [16], genetics [17,18], muscle nutrition [19], and peripheral blood cells [20], among other factors. However, there have only been a few studies on *S. sihama* diet, focusing on the appropriate protein and lipid levels [21], vitamin A, B, and C requirements [22,23,24], and the replacement of FM with dehulled soybean meal [25].

The liver is a frontline organ involved in regulating metabolic processes, hormone production, detoxification, and immune responses [26,27]. Liver disease and fatty liver have become stubborn problems plaguing intensive fish farming, and are often associated with an unbalanced nutrient mix in commercial diets, or the use of poor-quality diet ingredients [27]. In contrast to FM, plant protein limiting factors can cause metabolic disturbances, chronic inflammation, apoptosis, and immunosuppression in fish, ultimately leading to liver damage [28,29,30].

To date, there has been little work to map the regulatory mechanisms associated with the hepatic metabolic response of *S. sihama* fed a high-level low-gossypol cottonseed meal (LCSM) diet. The aim of this study was to investigate the potential mechanisms of metabolic regulation of growth performance, liver health, and the liver transcriptome in *S. sihama* fed a normal FM or high-level PP diet (Figure 1).

## 2. Material and Methods

### 2.1. Feed Ingredients

The proximate composition and amino acid composition of fishmeal (FM) and LCSM are shown in Table 1.

### 2.2. Experimental Design and Diet Preparation

Table 2 lists the nutritional makeup and composition of the experimental diets. Two isonitrogenous, isolipidic diets were created, replacing 0% and 64% of fishmeal with LCSM-equivalent amounts of protein (FM and R64). To correct the imbalance, methionine and lysine were given to the experimental diets. After being crushed, all raw materials were put through 60-mesh screens. In a Hobart-style mixer, feed ingredients were carefully weighed in accordance with the formula and thoroughly blended. Oils, choline chloride, and water were added before the complex was formed into pellets using an F-26 double-screw extruder. The diets had a moisture content of around 10% when they were air-dried, packaged in plastic Ziploc bags, and kept at −20 °C.

The average percentages of crude protein and lipid in the experimental diets were 49.74% and 9.10%, respectively, with no appreciable variation. Table 2 and Table 3 illustrate, respectively, the approximate makeup of the test diets and the amount of amino acids present.

### 2.3. Experimental Animals and Breeding Management

From Guangdong Ocean University in Zhanjiang, China, juvenile *S. sihama* were obtained. Prior to the trial, fish were given commercial diets and temporarily cultured in a cement pond that measured 4.5 m by 3.45 m by 1.8 m. The acclimatization period’s environment and breeding management were comparable to those in the official experiment [24]. A total of 180 juvenile, healthy *S. sihama* (initial body weight: 5.80 ± 0.58 g) were divided at random among 6 fiberglass tanks. Three tanks, each holding 30 fish, were given one of each diet at random. The experiment was conducted in a marine culture system indoors at the Guangdong Ocean University Marine Biology Research Base. For a 56-day testing period, the seawater’s salinity was altered to a range of 6 to 8. Prior to use, seawater was treated with chlorine dioxide and allowed to aerate for 24 h. For the first two weeks, the water was changed once weekly; for the following six weeks, it was changed twice weekly, with a 300 L water change between each. The pH ranged from 7.5 to 8.0 and the daily range of water temperature was 28.4–31.5 °C. The dissolved oxygen content was ≥6 mg/L, and the ammonia nitrogen and nitrite concentrations were ≤0.5 mg/L. Fish were fed twice daily, at 8:00 and 16:00, until they appeared to be fully satiated (at the beginning of feeding, fish fed at the surface of the water and feeding intensity subsequently decreased; when the fish swam towards the bottom of the tank and no longer came to feed, this was considered apparent saturation). The Guangdong Ocean University’s Animal Care and Use Committee approved all animal procedures, which were carried out in accordance with NIH guidelines.

### 2.4. Sample Collection and Analysis

At the termination of the feeding trial, fish were fasted for 24 h and anesthetized with eugenol (1:10,000; purity 99%, Shanghai Reagent, China) before sampling. Fish in each tank were counted and weighed to assess the survival rate (SR), weight gain rate (WGR), specific growth rate (SGR), and feed conversion ratio (FCR). For each treatment, three fish were chosen at random, and the livers were sectioned and fixed to provide histological sections. Following the final weighing, three fish were selected at random from each repeating tank, and their livers were rapidly removed and frozen in liquid nitrogen for use in the quantitative real-time PCR and transcriptome analyses that followed. Six fish were randomly selected from each dietary replicate, measured, and weighed for the purpose of calculating the condition factor (CF), and then frozen at −20 °C for proximate carcass analysis. According to the approved procedures of the Association of Official Analytical Chemists, the raw materials, dietary proximate, and whole-body proximate analyses of dry matter (dried at 105 °C), crude protein (by Kjeldahl apparatus, nitrogen 6.25), crude lipid (extraction with petroleum ether by Soxhlet apparatus), and ash content (incineration at 550 °C) were determined [31]. After being acid hydrolyzed in 6 M HCl for 24 h at 110 °C, the raw materials and dietary amino acid profile were analyzed by an automatic amino acid analyzer 433D (SYKAM) [14,32]. Free gossypol was determined according to GB/T 13086-2020

### 2.5. Histology of Liver

Histological analysis was performed according to the method described in a previous study [33]. The liver samples were preserved in 70% ethanol after being fixed in Bouin’s solution for 24 h. The materials were dehydrated in ethanol at varying concentrations before being set in paraffin wax. Hematoxylin–eosin (H&E)-stained sagittal slices of 5–7 μm thickness were produced for examination using a Nikon ECLIPSE 80i microscope (Nikon Corporation, Kanagawa, Japan).

### 2.6. Transcriptome Sequencing

The animal total RNA was extracted according to the instruction manual of the TRlzol Reagent (Life technologies, Carlsbad, CA, USA). RNA concentration and purity was measured using a NanoDrop 2000 (Thermo Fisher Scientific, Wilmington, DE, USA). RNA integrity was assessed using the RNA Nano 6000 Assay Kit of the Agilent Bioanalyzer 2100 system (Agilent Technologies, Santa Clara, CA, USA).

Based on the Illumina 2000 sequencing technology, a typical transcriptome cDNA library was built, and double-end sequenced using the Paired-End125 method to produce a large number of transcript sequences with a read length of 2 × 10 bp. Each sample received 4 Gb of transcriptome data.

### 2.7. De Novo Assembly of Sequencing Reads Raw

First, internal Perl scripts were used to process raw FASTQ data. In this stage, low-quality reads, adapter- and ploy-N-containing reads, and adapter-containing reads from the original data were removed to produce clean data. The clean data’s Q20, Q30, GC-content, and sequence duplication level were determined. The foundation for each and every downstream analysis was clear, high-quality data.

The Trinity package was used to perform de novo assembly on clean liver readings. In brief, the reads were assembled into distinct transcript sequences in Inchworm using greedy K-mer extension (K-mer = 25). Chrysalis inserted reads into de Bruijn graphs after mapping reads to Inchworm contigs. Finally, the Butterfly module processed each graph in parallel, producing full-length transcripts. This transcriptome shotgun assembly project has been registered in the SRA database as PRJNA766354.

### 2.8. Annotations of Transcripts and Pathways

BLASTX was used to search the assembled transcripts against the Nr (NCBI nonredundant protein sequences) [34], Nt (NCBI nucleotide sequences), and Swiss-Prot databases with E-values of 1.0 × 10^−5^ (E-values less than 1.0 × 10^−5^ were considered significant). Using the RPS-BLAST tool from the locally installed NCBI BLAST + v2.2.28 and HMMER 3.0 programs, domain-based comparisons with the Pfam (protein family) and KOG (a eukaryote-specific variant of the clusters of eukaryotic ortholog groups) databases were conducted.

Each transcript was initially allocated the top gene identifications and names. Transcripts were also used to identify the gene ontology (GO) term and the Kyoto Encyclopedia of Genes and Genomes (KEGG) pathway. Blast 2GO was used to achieve GO enrichment (biological process, cellular component, and molecular function at Level 2) [35].

The overview of metabolic pathway analysis was performed using online KEGG (http://www.genome.jp/kegg/, accessed on 26 July 2020), which is a database [36] resource for understanding high-level functions and utilities of the biological system.

### 2.9. Gene Set Enrichment Analysis (GSEA)

We used the GSEA and MSigDB [37] tools to perform gene set enrichment analysis to determine if a set of genes in specific pathways differed significantly between two groups. To summarize, we input a gene expression matrix and ranked genes using the signal-to-noise normalization method. The enrichment scores and *p*-value were computed using the default parameters.

### 2.10. Quantitative Real-Time PCR

The liver’s total RNA was extracted using a General RNA Extraction Kit, and its integrity was determined by electrophoresis on 1.2% denatured agarose gel. A NanoDrop 1000 spectrophotometer was used to test RNA integrity and quality. Following the manufacturer’s instructions, a PrimeScript^TM^ RT-PCR Kit was used to perform first-strand cDNA synthesis in RT [38]. The RNA was processed with gDNA Eraser before reverse transcription using a PrimeScript RT Reagent Kit. In a quantitative thermal cycler, real-time PCR tests were performed in a 10 μL reaction volume containing 5 μL SYBR@ Green. Real-time PCR Master Mix, 1 μL cDNA, 0.8 M primers, and 3.2 μL sterile double-distilled water. Each sample in the reaction was treated three times. Data acquisition occurred every 6 s, thus the thermal programmer was set to 30 s at 95 °C, 40 cycles at 95 °C for 5 s, 60 °C for 34 s, and a melt curve step from 60 °C to 95 °C at a rate of 0.5 °C/s. Based on the results of our preliminary experiment evaluating internal control genes, 60S ribosomal protein L38 was employed as a reference gene to normalize cDNA loading. Results were presented as mean ± standard error (standard deviation of the mean, SD). SPSS 19.0 software (SPSS Inc., Chicago, IL, USA) was used to analyze data by t-test and to confirm differences between the control and the R64 groups. The difference was considered statistically significant when *p* < 0.05.

### 2.11. Formula and Statistical Analysis

The following parameters and indices were calculated using a standard formula [2]: growth performance parameters, including WGR, SGR, and SR; feed utilization indices, including FCR, PER, and daily feeding intake rate (DFI); and morphology indices, including CF, hepatosomatic index (HSI), and viscerosomatic index (VSI). Independent-samples T-test was performed to evaluate the significant differences among measured variables between the FM diet and R64 diet (*p* < 0.05). All data analyses were conducted using the SPSS Statistics 19.0 software (SPSS, Michigan Avenue, Chicago, IL, USA).

## 3. Results

### 3.1. Growth Performance

As shown in Figure 2, the R64 diet had significantly lower final body weight (FBW), WGR, SGR, protein efficiency ratio (PER), and condition factor (CF) than the FM diet (*p* < 0.05), while there were no significant differences in FCR, VSI, HSI, SR, and DFI between the two groups (*p* < 0.05).

### 3.2. Whole-Body Proximate Composition

The results of whole-body proximate composition are shown in Figure 3. By replacing FM with LCSM, whole-body moisture, crude lipid, and ash were not significantly affected (*p* > 0.05), while only whole-body crude protein decreased significantly (*p* < 0.05).

### 3.3. Histological Observation of Liver

The control group (FM group) had normal liver structure with clear, intact, and well-arranged hepatocytes, and round nuclei of hepatocytes located in the center of the cytoplasm (Figure 4 FM). However, the liver of the experimental group (R64) showed several abnormalities, including diffuse lipid vacuolization, hepatocyte hypertrophy, nuclear consolidation, irregular arrangement of hepatocytes and central venous endothelial cells, inconspicuous hepatic cords, and hepatic sinusoidal stenosis (Figure 4 R64).

### 3.4. De novo Assembly for S. sihama

After de novo assembly analysis based on all Illumina clean reads, a total of 45,286 transcripts (ranging from 201 to 18,103 bp), with an N50 size of 2142 bp, were obtained for *S. sihama* (Table 4). For all six sequencing libraries, the percentage of reads that could be mapped to the assembled reference sequences was higher than 83.13% (Table 5).

### 3.5. Annotation and Function Analysis of Liver Transcripts

Transcripts were annotated and analyzed by comparison with the Nr, KEGG, COG, Swiss-Prot, and GO databases. The results showed that a total of 22,857 transcripts (50.47%) were annotated in at least one database, of which, 22,721 annotated transcripts (50.17%) had significant BLAST hits against the Nr database. The detailed annotation results are listed in Table 4.

Among the top-hit species matched with the Nr database, 8120 Unigene sequences were similar to *Larimichthys crocea*, followed by *Lates calcarifer* (5665), *Stegastes partitus* (2102), *Paralichthys olivaceus* (897), *Oreochromis niloticus* (611), and others (5326) (Figure 5).

### 3.6. Transcriptome Differential Expression Analysis

The differential expression (DE) analysis was performed between the R64 group and the FM group. A total of 166 DEGs were identified, of which 45 and 122 were up- and downregulated, respectively. The top 20 genes up- and downregulated in the R64 group are represented as a heat map (Figure 6).

Of the top 20 differentially expressed genes, genes such as *cyp51* (a member of the cytochrome P450 enzyme superfamily), *hsp70* (heat shock protein 70), *il6st* (interleukin 6 cytokine), and *bcl6b* were positively regulated in the R64 group. In addition, the R64 group showed significant downregulation in metabolism-related genes such as *arg2* (catalyzes hydrolysis of arginine to ornithine and urea), *csrp2*, *hp* (haptoglobin), *sdhb* (involved in the oxidation of succinate), and *nr1d1* (lipid and bile acid metabolism, adipogenesis) (Figure 6).

### 3.7. Function Enrichment and Classification

All transcripts in this study were functionally enriched and taxonomically analyzed by GO and KEGG (Figure 7A,B). GO annotation provides a controlled vocabulary to describe gene function. By matching the GO database, 14,356 (31.70%) single genes were annotated with 26 terms. Among the various categories of biological processes, metabolic processes (down 29, up 9) were the most dominant group, followed by single-organism processes (up 23, down 8) and cellular processes (up 23, down 7). Within the molecular function category, catalytic activity (up 25, down 7) and binding (up 22, down 10) were the most dominant group. In terms of cellular component, 14 single genes were categorized as cell and cell part, followed by membrane (up 10, down 3) and membrane part (up 9, down 3) (Figure 7A).

The KEGG Orthology (KO) system is a classification system of proteins (enzymes) with highly similar sequences and similar functions in the same pathway. KEGG is categorized into a group of proteins and then labeled with KO. BLAST searches against the KEGG database revealed that a total of 21,919 (48.40%) single genes were assigned to 186 known pathways (Figure 7B). The number of sequences annotated in the KEGG pathways ranged from 8 to 1596. In metabolic mainly global overview maps, lipid metabolism and amino acid metabolism are included; the package in organismal systems mainly annotates the digestive system, the endocrine system, and the immune system. These annotations provide valuable information for the study of specific biological and metabolic processes, functions, and molecular mechanisms of fish in subsequent studies (Figure 7B).

### 3.8. Gene Set Enrichment Analyses (GSEA)

To fully understand the mRNA in the differential transcriptome, we analyzed the mRNA by gene set enrichment analysis (GSEA). The results showed that there were 15 enrichment pathways in the downregulated mRNAs, including DNA replication, arginine biosynthesis, tyrosine metabolism, proximal tubule bicarbonate reclamation, alanine, aspartate, and glutamate metabolism, etc. (Table 6). The upregulated mRNAs enrichment pathways include 10 pathways, involving nasal biosynthesis, B cell receptor signaling pathway, glycosphingolipid biosynthesis-lacto, neolacto series and Notch signaling pathway, sphingolipid signaling pathway, Hippo signaling pathway of multiple species, etc. (Table 7).

### 3.9. Gene Expression (qPCR)

To validate the RNA-seq data, we selected ten genes from the differential genes (Figure 8). Only the qPCR result for *tjp1* was opposite to that of RNA-seq, while the remaining nine genes all showed the same trend as RNA-seq, confirming the reliability of the RNA-seq results.

## 4. Discussion

Low-gossypol cottonseed meal has been employed as a high-quality protein source in grass carp (*Ctenopharyngodon idellus*) [39], crucian carp (*Carassius auratus*) [40], black sea bass (*Centropristis striata*) [41], and blunt snout bream (*Megalobrama amblycephala*) [42]. In the current investigation, we discovered that high amounts of LCSM in the diet had a negative effect on *S. sihama* growth indicators. The decrease in growth indicators was most likely caused by a reduction in dietary intake. In the studies of blunt snout bream [42] and black sea bass [41], it was shown that high levels of PP in the diet reduced the feeding of fish. Previous research suggested that various levels of PP substituting fishmeal in the diet may have induced changes in body composition on a wet matter basis [39,40,43,44]. In this study, the whole-body crude protein content of fish given the FM diet was significantly higher than that of fish fed the R64 diet. The LCSM replacement fishmeal had no effect on whole-body crude lipid levels. Because this meal was less appealing and was consumed at a much lower rate than the FM diet, the comparatively low whole-body crude protein levels in fish fed the high level of LCSM may indicate greater protein catabolism with less protein storage. Fish fed the R64 diet had 72.58% moisture, while fish fed the FM diet had 71.32% moisture (Figure 3a). These moisture content changes were also likely attributable to decreased palatability and consumption of the high LCSM diet with less protein storage, resulting in a proportional increase in moisture content while ash levels remained the same. Experiments with PP to substitute fishmeal yielded comparable effects in hybrid snakehead (*Channa maculata* ♀ × *Channa argus* ♂) [45], cobia (*Rachycentron canadum*) [46], red sea bream (*Pagrus major*) [47], and silver crucian carp (*Carassius auratus gibelio* ♀ × *Cyprinus carpio* ♂) [48].

Histological changes are an important aspect in understanding the pathological alterations associated with the nutritional origin of the fish. The replacement of fishmeal by LCSM in the diet led to a number of histopathological changes in *S. sihama*’s liver, including diffuse lipid vacuolation, hepatocyte hypertrophy, nuclear consolidation, irregular arrangement of hepatocytes and central venous endothelial cells, obscure hepatic cords, and sinusoidal stenosis. This means that, although some of the antinutritional factors have been removed after cottonseed protein has been degossypolized, the PP itself contains high levels of fiber, polysaccharides, and nonfat soluble antinutritional factors, which may still affect the health of the fish liver. Studies in grass carp [49], Asian seabass (*Lates calcarifer*) [50], and hybrid sturgeon (*Acipenser schrenckii* ♀ × *A. baeri* ♂) [51] have also found that high levels of PP in the diet can cause negative effects on the liver.

The effects of nutrition on the liver transcriptome of tilapia (*Oreochromis niloticus*) [52], Atlantic salmon (*Salmo salar*) [53], yellow perch (*Perca flavescens*) [54], and pearl gentian grouper (*Epinephelus fuscoguttatus* ♀ *× Epinephelus lanceolatus* ♂) [55] have been well studied. However, the effects of nutrition on the liver transcriptome of *S. sihama* have not been reported. The dietary transcriptional profile of nutritional stress (induced by very high LCSM) includes downregulation of hepatic metabolic pathways, including genes related to amino acid metabolism, glycerolipid metabolism, and the metabolism of xenobiotics. Upregulation includes pathways related to lipid metabolism and immune response. Interestingly, this profile is similar to that observed in pearl gentian grouper fed high levels of soybean meal [55].

Bioinformatic analysis of the liver transcriptome of *S. sihama* fed FM or R64 diets showed that LCSM had the greatest effect on metabolic and immune-related genes in the liver. In the GO term enrichment differential analysis, 29 metabolic process differential genes were downregulated and 9 genes upregulated. In the KEGG enrichment analysis, regarding metabolism pathways, 73 differential genes were found, most of which were concentrated in global and overview maps (30), lipid metabolism (12), amino acid metabolism (10), and carbohydrate metabolism (7). In the GSEA, the R64 group showed downregulation of some pathways of amino acid metabolism, glucose metabolism, and lipid metabolism compared to the FM group.

A high proportion of PP in the diet in place of fishmeal is thought to be detrimental to protein synthesis. This has been demonstrated in studies with starry flounder (*Platichthys stellatus*) [56], turbot (*Scophthalmus maximus* L.) [57], and rainbow trout (*Oncorhynchus mykiss*) [58]. In the present study, the substitution of FM by LCSM in the diet by GSEA was found to have resulted in downregulation of arginine biosynthesis, tyrosine metabolism, alanine, aspartate, and glutamate metabolism, glutathione metabolism, phenylalanine metabolism, cysteine and methionine metabolism, and tryptophan metabolic pathways. The decrease in amino acid metabolic pathways may be one of the reasons for the lower growth indicators in the R64 group compared to the FM group.

Combining GSEA and KEGG enrichment analysis, we looked for the following candidate genes for amino acid metabolism: *arg2*, *glud1*, *ddc*, *fah*, *gstz1*, *adsl*, *cbs*, and *cth*.

Arginase 2 (*arg2*) is generally thought to produce ornithine as a precursor for polyamine, glutamate, and proline biosynthesis [59], synthesize urea for osmoregulation [60,61], and regulate arginine levels required for nitric oxide (NO) synthesis [62]. In the present study, LCSM substituted for FM had a significant inhibitory effect on *arg2* expression, which may have been caused by the greater content of arginine in LCSM than in FM. There are a number of reports indicating that arginine, in addition to its immune-enhancing effects, can also mediate immunosuppressive effects when added in excess to the diet [63,64,65]. In a recent study, it was also found that supplementation of arginine in the diet may impair the cell-mediated immune response of sea bass to some extent, reducing the number of circulating neutrophils and monocytes [65]. DOPA decarboxylase (*ddc*) is a 5′-phosphate pyridoxal (PLP)-dependent enzyme, known as aromatic 1-amino acid decarboxylase, catalyzes L-3, 4-dihydroxyphenylalanine (L-DOPA), and 5-hydroxytryptophan (5-HTP), respectively. It plays a key role in biological behavior, development, and parasite defense [66]. In this study, the expression level of *dcc* was significantly downregulated in the R64 diet, which may affect the behavior, development and parasite defense of *S. sihama*. *Ddc* has been found to be responsible for controlling dopamine synthesis in shrimp (*Litopenaeus vannamei)*, and then regulating physiological and immune responses in shrimp [67]. In addition, studies on *Drosophila melanogaster* also showed that *dcc* was closely related to immunity [68].

Glutamate dehydrogenase 1 (GLUD) is a crucial enzyme in glutamine decomposition, converting glutamate to alpha-ketoglutaric acid (α-kg) to enter the TCA cycle, where NAD(P)^+^ is reduced to NAD(P)H. *Glud* is triggered by the direct binding of the essential amino acid leucine, which increases glutamate deamination and the formation of α-kg. The GLUD isoenzymes GLUD1 and GLUD2 are both increased in human malignancies, allowing cancer cells to employ this pathway for growth and proliferation [69]. GLUD1 is not only required for cancer cells to maintain the TCA cycle in order to accomplish rapid proliferation and development, but it can also activate mTORC1 [70]. Adenylate succinate lyase (ADSL) is an essential enzyme involved in the de novo biosynthesis of purine. ADSL participates in the purine nucleotide cycle, together with adenylate succinate synthase and AMP deaminase. This cycle helps to rigidly regulate cellular metabolism by providing the number of available Krebs cycle intermediates, and pulling the adenosine kinase reaction in the direction of AMP formation. Therefore, ADSL plays an important role in both cell replication and metabolism [71]. In this study, the expression levels of *glud* and *adls* in the livers of the R64 diet group were significantly decreased, indicating that LCSM may affect cell proliferation, purine metabolism, and cell replication in *S. sihama* liver, thus affecting the growth and development of *S. sihama*, which was also consistent with the fact that the growth indices in the R64 diet group were significantly lower than those in the FM diet group.

Cystathionine beta-synthase (CBS)-catalyzed transsulfuration converts homocysteine to cystathionine, which is converted to cysteine by cystathionine γ lyase. Additionally, CBS converts homocysteine to methionine (remethylation) or to taurine (desulfuration). This catabolic pathway alleviates the toxicity of methionine by removing excess homocysteine [72]. In the present experiment, the R64 group *cbs* expression levels were significantly lower than the FM group, suggesting that methylation was downregulated in the livers of fish fed the FM diet, and that the conversion pathway from methionine to taurine may be attenuated in the LCSM treated group. Taurine addition to the diet has been shown to improve growth in rainbow trout [73], red sea bream [74], Japanese flounder (*Paralichthys olivaceus*), and common carp [75]. In some fish, taurine can be synthesized from methionine via cystine. However, at high levels of PP in the diet, the rate of taurine synthesis may not be sufficient to meet the requirements of the species, thus, taurine is conditionally essential [74,75]. In addition, the cystathionine gamma-lyase (CTH) gene was also significantly downregulated in the R64 group. CTH catalyzes the final step in the transsulfuration pathway from methionine to cysteine, converting cystathionine to cysteine, ammonia, and 2-oxobutyric acid; glutathione synthesis in the liver is dependent on the availability of cysteine. However, the results of this experiment suggest that LCSM may affect hepatic glutathione synthesis by downregulating *cbs* and *cth*. Heat shock protein family 70 (*hsp70*) is involved in a variety of cellular processes as a molecular chaperone, maintaining protein homeostasis during cellular stress through two opposing mechanisms: protein refolding and degradation. *Hsp70* is strongly upregulated by heat stress and toxic chemicals [76]. In this experiment, substitution of fishmeal by LCSM caused the high expression of *hsp70* in the liver of *S. sihama*, indicating that although the degossyphenols of cottonseed protein had been carried out, some other antinutritional or toxic substances in LCSM would still cause metabolic stress and damage to the liver of *S. sihama*. Fumaryl acetoacetic acid hydrolase (FAH) is responsible for the final step of tyrosine metabolism. Reduced expression of *fah* leads to accumulation of the toxic metabolite fumaracetic acid (FAA) in hepatocytes and proximal renal tubules, leading to mutation and apoptosis [77]. In this experiment, *fah* expression levels were significantly reduced in the R64 diet group, suggesting that LCSM substitution for FM may cause apoptosis in *S. sihama* hepatocytes, as corroborated by H&E sections of the liver. Studies in pigs have found that FAH deficiency leads to diffuse and severe hepatocyte damage [78]. The glutathione S-transferase zeta 1 (GSTZ1) gene belongs to the glutathione S-transferase (GST) superfamily, which encodes multifunctional enzymes that detoxify electrophilic compounds, such as carcinogens, mutagens, and various medicinal drugs, by binding to glutathione. One of the steps in the phenylalanine/tyrosine degradation pathway is catalyzed by this enzyme, which converts maleyl acetoacetate to fumaryl acetoacetate. Oxidative stress is caused by the absence of a comparable gene in mice [79]. *Gstz1* expression levels were significantly lower in the R64 group than in the FM group in this study, suggesting that LCSM may have an oxidative stress effect on the liver. In summary, high levels of LCSM in the diet in place of fishmeal can cause oxidative stress in the liver of *S. sihama*. Studies on silvery-black porgy (*Sparidentex hasta*) [80], turbot [76], and Ussuri catfish (*Pseudobagrus ussuriensis*) [43] have also reached a similar conclusion, that adding high levels of plant proteins to the diet can cause metabolic stress and damage of the liver.

PP replacement for fishmeal has been reported to affect lipid metabolism in trout [81], Atlantic salmon (*Salmo salar*) [82], and hybrid grouper (*Epinephelus fuscoguttatus* ♀ × *E. lanceolatus* ♂) [83]. In this experiment, we found that the triglyceride metabolic pathway appeared to be significantly downregulated in the R64 diet by GSEA, whereas the steroid biosynthetic pathway appeared to be significantly upregulated in the R64 group compared with the FM group. Comprehensive GSEA and KEGG enrichment analysis identified the following candidate genes: *pnpla2*, *mogat2-a*, *lipg*, *cyp51*, *sqle*, *sc5d*, *lss*, and *soat1*.

Patatin-like phospholipase domain-containing protein 2 (PNPLA2), which was designated adipose triglyceride lipase (ATGL) 17 years ago [84], is assumed to be the central rate-limiting enzyme for fatty acid mobilization in mammals [85]. PNPLA2 specifically hydrolyses triglycerides (TG) to diacylglycerols (DG) and releases free fatty acids (FFA). In this experiment, we found a significant downregulation of *pnpla2* in the R64 group, suggesting that LCSM substitution for fishmeal may lead to abnormal hepatic lipolysis in *S. sihama*. There is growing evidence that dysregulated lipolysis is associated with a number of metabolic disorders, such as fatty liver and type two diabetes [86]. LCSM replacement of fishmeal resulted in abnormal accumulation of liver lipid in *S. sihama*, as shown in Figure 4. Studies in zebrafish and mice have also found that similar lipid deposition is observed with inhibition of *pnpla2* [85,87]. In the present study, there was a significant downregulation of *mogat2* expression levels in the R64 diet, suggesting that LCSM affects lipid synthesis in addition to lipolysis. This may also explain the small difference in whole-body crude lipid content between the FM and R64 diets, and the lack of significant differences in VSI and HSI between the two groups. In fact, in a starry flounder study, it was also found that the substitution of PP for fishmeal resulted in a decrease in lipid synthesis and catabolism [12].

In addition, we found that *cyp51*, *sqle*, *sc5d*, *lss*, and *soat1* were all upregulated in this experiment, all of which are key genes in the cholesterol biosynthesis pathway, which means that the LCSM substitution for fishmeal actually promoted the cholesterol synthesis ability of *S. sihama*. In numerous studies in the past, PP substitution for fishmeal has been found to reduce serum cholesterol levels [88,89]. There is also an abundance of evidence that phytosterols in plant proteins can inhibit cholesterol absorption [90,91,92]. LCSM replaced fishmeal and reduced the content of cholesterol in the diet. In addition, dietary phytosterol contained in PP inhibited the absorption of cholesterol in fish, which led to the possibility that the cholesterol-satisfied growth requirement could not be obtained in the diet. Therefore, genes related to de novo cholesterol synthesis in the liver were upregulated. A study found that the growth of rainbow trout fed with added cholesterol in a PP-based diet was significantly higher than that in the nonsupplemented diet [93], and similar results were found in channel catfish (*Ictalurus punctatus*) [94], Japanese flounder [95], and turbot [96], which also supported our view.

The effects of fishmeal replacement by PP on glucose metabolism is rarely studied. In this study, GSEA showed that the mutual conversion pathway of pentose and glucuronate interconversions was significantly downregulated in the R64 group compared with the FM group. Based on GSEA and KEGG enrichment analysis, we found the following candidate differential genes: *eno1*, *fbp1*, *g6pd*, and *pgam2*.

ENO1 and PGAM2 are both important enzymes in glycolysis. ENO1 catalyzes the conversion of 2-phosphoglycerate to phosphoenolpyruvate, and PGAM2 catalyzes the reversible reaction from 3-phosphoglycerate (3-PGA) to 2-phosphoglycerate (2-PGA). In this study, LCSM replacement of fishmeal resulted in significant downregulation of *eno1* and *pgam2* in the liver, indicating that LCSM replacement of fishmeal could inhibit glycolysis. It may be that the residual gossypol in LCSM exerted an inhibitory effect on liver glycolysis. In vitro studies found that decreased sperm motility of rabbit could result from a reduced energy supply after inhibition of glycolysis by gossypol [97]. In addition, gossypol is thought to inhibit glycolysis by inhibiting lactate dehydrogenase isozyme 5 [98].

Fructose-bisphosphatase 1(FBP1) catalyzes the hydrolysis of fructose 1,6-bisphosphate to fructose 6-phosphate, acting as a rate-limiting enzyme in the process of gluconeogenesis. Glucose-6-phosphate dehydrogenase (G6PD) is the final step in gluconeogenesis. In this experiment, both *fbp1* and *g6pd* were significantly downregulated, indicating that fishmeal replacement by LCSM can cause inhibitory effects on *S. sihama* gluconeogenesis. This may be because the level of glycogenic amino acids in LCSM is less than that in the fishmeal diet. The total content of glycogenic amino acids in fishmeal is 472 g/kg, while the total content of glycogenic amino acids in LCSM is 441.2 g/kg (Table 1). The reduction of glycogenic substrates may be one reason for the inhibition of gluconeogenesis. In addition, the availability of amino acids in LCSM may be low. In silvery-black porgy, liver dysfunction is caused by loss of hepatocyte integrity, disrupted protein synthesis, and low amino acid availability [99]. In this experiment, we only supplemented methionine and lysine, while some glycogenic amino acids, such as alanine and aspartic acid, in the LCSM diet were less than the FM diet. In a Totoaba (*Totoaba macdonaldi*) study [100], it was found that more than 45% soybean meal in the diet affected the FBPase activity in the liver of Totoaba. Similar results were also reported in rainbow trout, where lower FBPase activity was associated with lower alanine content in the diet [101]. Therefore, we believe that FBPase activity is limited by gluconeogenesis, which can also be attributed to low protein digestibility [102], which may affect alanine availability [103]. In addition, a turbot study also found that adding high levels of soybean meal and cottonseed meal to the diet also inhibited G6PD activity, and thus gluconeogenesis [104]. These effects may be caused by the inefficient use of PP by carnivorous fish.

## 5. Conclusions

In conclusion, this study highlights the negative effects of substituting fishmeal with low-gossypol cottonseed meal (LCSM) in the diet of juvenile *Sillago sihama*. The growth performance of the fish was significantly lower when fed a formulated diet with 64% LCSM compared to a traditional fishmeal-based diet. Additionally, the liver of *S. sihama* fed the LCSM diet showed diffuse lipid vacuolation, indicating harm to the liver’s structural integrity. The liver transcriptome analysis revealed that the LCSM diet negatively impacted lipid synthesis and catabolism, pathways for cholesterol synthesis, amino acid metabolic pathways, as well as hepatic gluconeogenesis and glycolysis. Therefore, this study suggests that it is not recommended to substitute high levels of LCSM for fishmeal in the diet of *S. sihama*.

## Figures and Tables

**Figure 1 animals-13-01194-f001:**
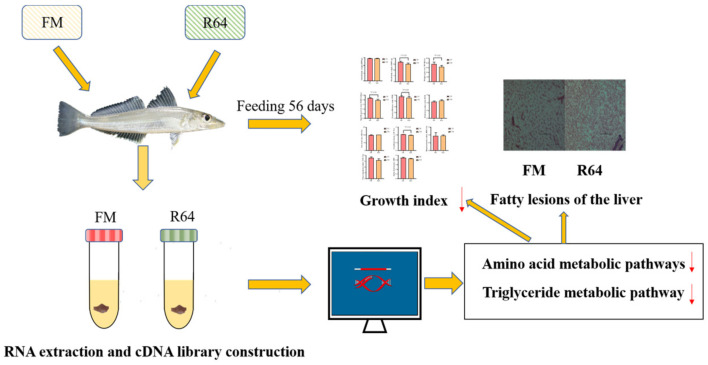
The experimental protocol and result of this study. The red arrow indicates suppression.

**Figure 2 animals-13-01194-f002:**
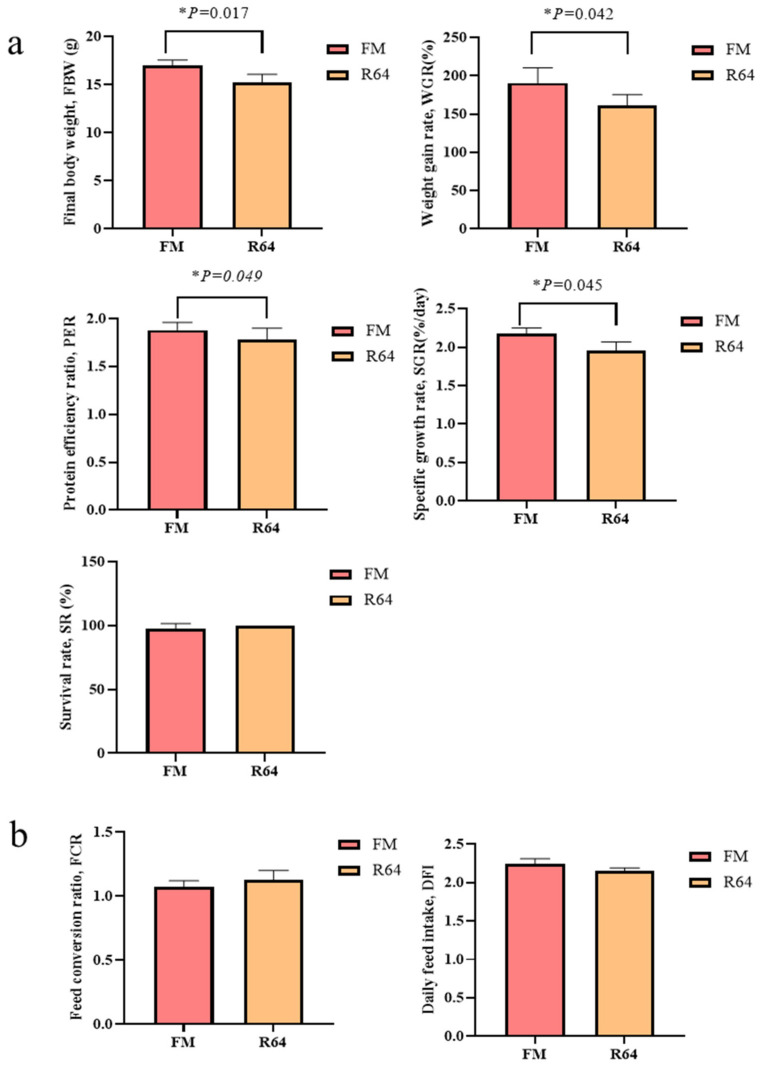
Effects of fishmeal replacement by low-gossypol cottonseed meal on the growth performance of juvenile *S. sihama.* (**a**): growth index, (**b**): feed utilization, (**c**): morphological indices.

**Figure 3 animals-13-01194-f003:**
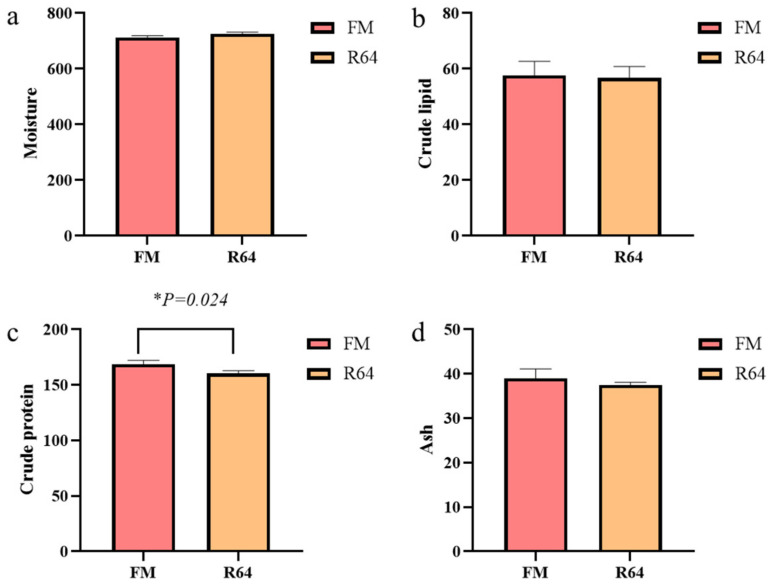
Effects of fishmeal replacement by low-gossypol cottonseed meal on whole-body proximate composition of juvenile *S. sihama* (g/kg wet matter). (**a**): Moisture, (**b**): Crude lipid, (**c**): Crude protein, (**d**): Ash.

**Figure 4 animals-13-01194-f004:**
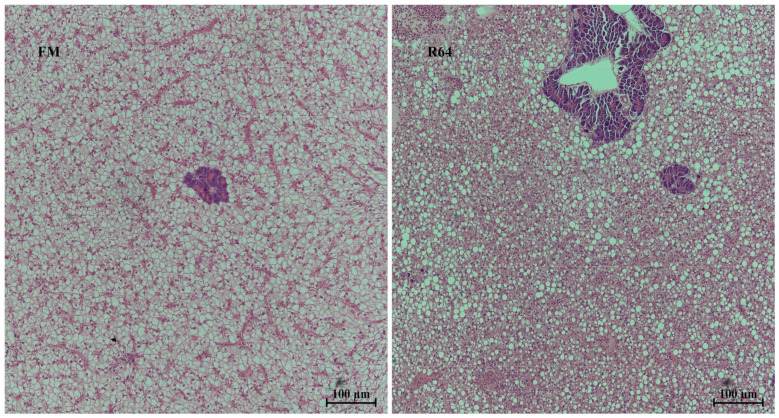
Liver histology (magnification 200×) of juvenile *S. sihama*.

**Figure 5 animals-13-01194-f005:**
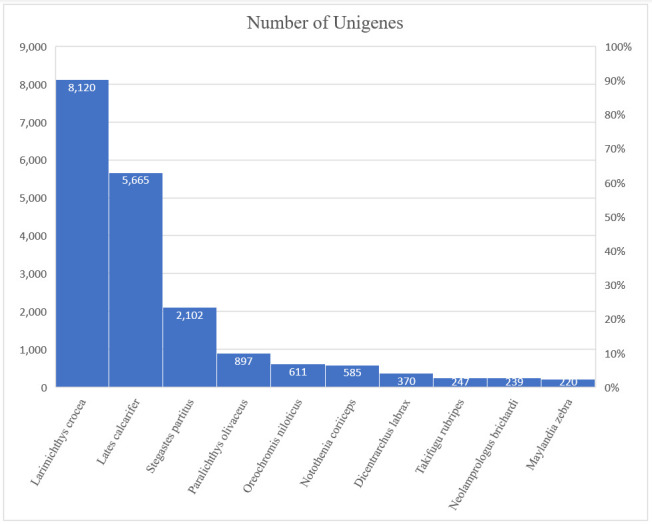
Species distribution results of a similarity search of Unigenes against the Nr database.

**Figure 6 animals-13-01194-f006:**
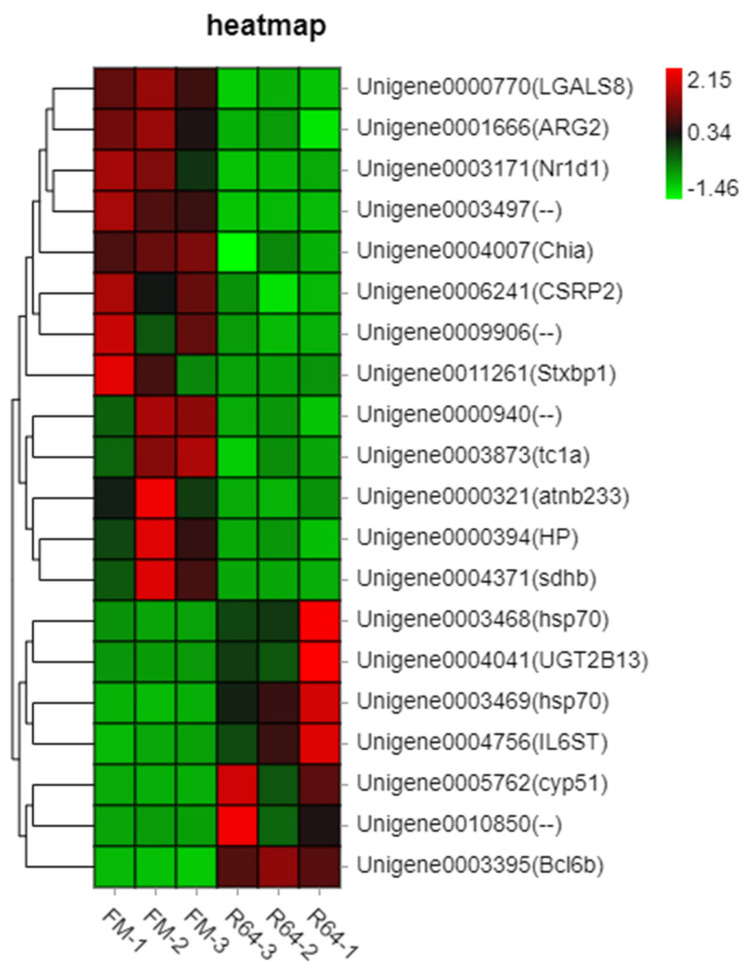
Heatmap of the top twenty genes differently expressed in liver of the FM group and R64 group. The graph was constructed in an R environment using the heatmap package, and using the normalized log2CPM (counts per million) as input.

**Figure 7 animals-13-01194-f007:**
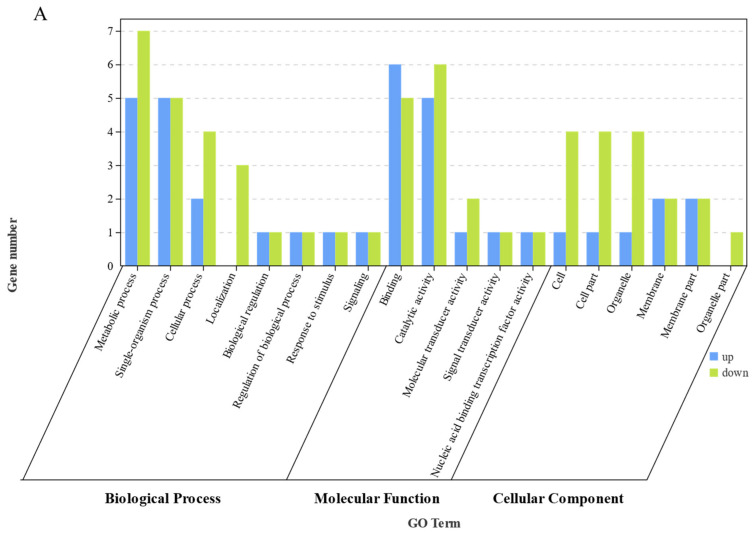
The Gene ontology (GO) enrichment analysis of differentially expressed genes (DEGs) in FM vs. R64 (**A**), and KEGG pathways of differentially expressed genes (DEGs) in FM vs. R64 (**B**).

**Figure 8 animals-13-01194-f008:**
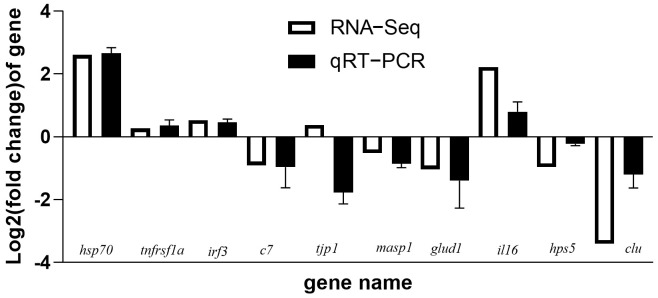
Comparison of mRNA expression changes between RNA−Seq and RT−qPCR of *S. sihama* liver. Fold change shows the expression level of mRNA in the R64 group compared with the FM group (n = 3). The relative expression values were normalized to EF−1 Alpha gene expression. Error bars indicate standard deviation.

**Table 1 animals-13-01194-t001:** Chemical composition of the test ingredients (g/kg dry matter).

Components	White Fishmeal (FM)	Low-Gossypol Cottonseed Meal (LCSM)
Dry matter	951.0	929.0
Crude protein	680.0	649.0
Crude lipid	73.0	5.0
Glycogenic amino acids (g/kg)	472.0	441.2
Free gossypol		0.079
NEAA ^a^		
Ala	40.3	22.9
Asp	62.4	54.2
Tyr	24.0	17.4
Ser	30.4	24.8
Glu	91.9	123.3
Gly	46.3	24.9
Cys	6.2	11.0
Pro	29.4	21.3
EAA ^b^		
Val	31.8	25.4
Met	19.9	7.4
Ile	27.7	18.3
Leu	49.3	34.3
Thr	29.2	18.7
Phe	26.0	33.1
His	13.5	16.8
Lys	50.7	24.7
Arg	43.0	72.2

^a^ NEAA: nonessential amino acid; ^b^ EAA: essential amino acid.

**Table 2 animals-13-01194-t002:** Ingredient composition and nutrient content of experimental diets (g/kg dry matter).

Ingredients	FM	R64
White fishmeal	550.0	196.0
Low-gossypol cottonseed meal	0.0	354.0
Soybean protein concentrated	50.0	50.0
Vital wheat gluten	70.0	70.0
High protein flour	189.5	189.5
Fish oil ^a^	21.9	44.8
Phospholipid	30.0	30.0
Mineral mixture ^b^	5.0	5.0
Vitamin mixture ^c^	2.0	2.0
Ca(H_2_PO_4_)_2_	15.0	15.0
Antioxidant	0.3	0.3
Choline chloride	5.0	5.0
Microcrystalline cellulose	50.3	13.9
Methionine ^d^	0.00	9.2
Lysine ^d^	0.00	4.3
Carboxymethyl cellulose sodium	10.0	10.0
Attractant ^e^	1.0	1.0
Proximate composition ^f^		
Moisture	116.3	122.9
Crude protein	495.6	499.2
Crude lipid	89.7	92.3
Ash	147.9	94.6
Free gossypol content (mg/kg)	0	27.96

^a^ Semirefined fish oil, Oleaginosa Victoria S.A., Peru. ^b^ Mineral mixture (mg/kg diet): KIO_4_ 0.15, CoCl_2_·6H_2_O 20.27, CuSO_4_·5H_2_O 99.2, FeC_6_H_5_O_7_ 68.55, ZnSO_4_·7H_2_O 141.4, MgSO_4_·7H_2_O 0.6, CaH_2_PO_4_ 400, KCl 76.65, Na_2_SeO_3_ 10, eolite power 4183.1, (obtained from Zhanjiang Yuehai Feed Co. Ltd., Guangdong, China). ^c^ Vitamin mixture (mg/kg diet): vitamin B_1_, 51; vitamin B_2_, 50; vitamin B_6_, 100; vitamin B_12_, 0.2; vitamin K_3_, 10; vitamin E, 198; vitamin A, 20; vitamin D_3_, 100; nicotinic acid, 202; D-calcium pantothenate, 122; biotin, 50; folic acid, 12.5; inositol, 306.12; cellulose, 778.18, (obtained from Zhanjiang Yuehai Feed Co. Ltd., Guangdong, China). ^d^ Methionine and lysine were added to balance amino acids with the control group. ^e^ Attractant composition: taurine:glycine:betaine = 1:3:3, obtained from Hangzhou King Techina Technology (Hangzhou, China). ^f^ Moisture, crude protein, crude lipid, and ash contents were measured values.

**Table 3 animals-13-01194-t003:** Amino acid compositions of experimental diets (g/kg dry matter).

Amino Acid	FM	R64
NEAA ^a^		
Ala	25.56	19.40
Asp	40.65	37.75
Tyr	16.90	14.56
Ser	21.26	19.28
Glu	82.09	93.21
Gly	29.29	21.71
Cys	5.60	7.30
Pro	26.16	23.29
EAA ^b^		
Val	21.88	19.61
Met	12.25	17.02
Ile	19.58	16.25
Leu	33.61	28.30
Thr	19.05	15.33
Phe	19.63	22.15
His	9.90	11.07
Lys	31.30	26.39
Arg	28.89	39.23

^a^ NEAA: nonessential amino acid; ^b^ EAA: essential amino acid.

**Table 4 animals-13-01194-t004:** Summary of assembly and annotation statistics of the liver transcriptome of *S. sihama*.

Category	Number of Transcripts
Total number of clean reads of FM	130,402,786
Total number of clean reads of R64	214,044,798
N50 length of all transcripts (bp)	2142
N50 number of all transcripts (bp)	7212
Average length of all transcripts (bp)	1140
Max length (bp)	18,103
Min length (bp)	201
Total number of Unigenes	45,286
Total number of annotated transcripts in Nr database	22,721 (50.17%)
Total number of annotated transcripts in KEGG database	21,919 (48.40%)
Total number of annotated transcripts in COG database	14,356 (31.70%)
Total number of annotated transcripts in Swiss-Prot database	18,163 (40.10%)
Total number of annotated transcripts in at least one database	22,857 (50.47%)

**Table 5 animals-13-01194-t005:** Summary of statistics for Illumina short reads of the liver transcriptome of *S. sihama*.

Sample ^a^	Raw Reads	Clean Reads (%)	Q20(%) ^b^	Q30(%) ^c^	Total Mapped (%) ^d^
FM-1	42,421,754	42,377,494 (99.90%)	6,194,835,989 (97.77%)	5,930,956,341 (93.60%)	35,255,013 (83.19%)
FM-2	45,229,088	45,132,640 (99.79%)	6,540,079,726 (97.11%)	6,212,540,222 (92.25%)	37,518,053 (83.13%)
FM-3	42,980,934	42,892,652 (99.79%)	6,540,079,726 (97.11%)	5,929,765,641 (92.44%)	36,250,721 (84.51%)
R64-1	72,558,876	72,447,678 (99.85%)	10,555,569,968 (97.43%)	10,063,911,626 (92.90%)	61,309,373 (84.63%)
R64-2	84,672,696	84,469,722 (99.76%)	12,245,275,212 (97.03%)	11,601,802,583 (91.93%)	71,452,489 (84.59%)
R64-3	57,255,106	57,127,398 (99.78%)	8,288,929,038 (97.15%)	7,864,355,080 (92.18%)	48,512,646 (84.92%)
Total	345,118,454	344,447,584			

^a^ 1, 2, and 3: three independent biological replicates; ^b^ Q20: the percentage of bases with a Phred value > 20; ^c^ Q30: the percentage of bases with a Phred value > 30; ^d^ the number of clean reads that mapped onto the assembled reference transcriptome.

**Table 6 animals-13-01194-t006:** Fifteen KEGG-enriched pathways significantly downregulated in gene set enrichment analysis (GSEA).

id	Name	KEGG_A_Class	KEGG_B_Class	SIZE	ES	NES	NOM p-val	FDR q-val
KO03030	DNA replication	Genetic Information Processing	Replication and repair	47	−0.55602	−2.27893	0	0
KO00220	Arginine biosynthesis	Metabolism	Amino acid metabolism	25	−0.60027	−2.2081	0	0.001113
KO00350	Tyrosine metabolism	Metabolism	Amino acid metabolism	36	−0.54038	−2.13607	0	0.001418
KO04964	Proximal tubule bicarbonate reclamation	Organismal Systems	Excretory system	36	−0.51215	−2.06006	0	0.003754
KO00250	Alanine, aspartate and glutamate metabolism	Metabolism	Amino acid metabolism	48	−0.47441	−2.04144	0	0.003003
KO00790	Folate biosynthesis	Metabolism	Metabolism of cofactors and vitamins	31	−0.53424	−2.01373	0	0.003464
KO00040	Pentose and glucuronate interconversions	Metabolism	Carbohydrate metabolism	37	−0.49047	−1.92725	0	0.00808
KO00480	Glutathione metabolism	Metabolism	Metabolism of other amino acids	62	−0.42324	−1.84426	0	0.013533
KO00360	Phenylalanine metabolism	Metabolism	Amino acid metabolism	19	−0.5535	−1.80131	0	0.018464
KO00980	Metabolism of xenobiotics by cytochrome P450	Metabolism	Xenobiotics biodegradation and metabolism	47	−0.41641	−1.77253	0.006289	0.02113
KO00270	Cysteine and methionine metabolism	Metabolism	Amino acid metabolism	63	−0.39334	−1.75983	0	0.021907
KO00240	Pyrimidine metabolism	Metabolism	Nucleotide metabolism	116	−0.34261	−1.69487	0	0.030524
KO00380	Tryptophan metabolism	Metabolism	Amino acid metabolism	44	−0.39964	−1.6907	0.011299	0.026495
KO00561	Glycerolipid metabolism	Metabolism	Lipid metabolism	86	−0.36026	−1.67864	0	0.027124
KO00760	Nicotinate and nicotinamide metabolism	Metabolism	Metabolism of cofactors and vitamins	43	−0.39243	−1.62425	0.011364	0.033254

Note: ES: enrichment score; NES: normalized enrichment score; NOM p-val: nominal *p*-value; FDR q-val: false discovery rate.

**Table 7 animals-13-01194-t007:** Ten KEGG-enriched pathways significantly upregulated in gene set enrichment analysis (GSEA).

id	Name	KEGG_A_Class	KEGG_B_Class	SIZE	ES	NES	NOM p-val	FDR q-val
KO00100	Steroid biosynthesis	Metabolism	Lipid metabolism	23	0.602897	1.754509	0	0.037575
KO04662	B cell receptor signaling pathway	Organismal Systems	Immune system	125	0.431704	1.641614	0	0.095788
KO00601	Glycosphingolipid biosynthesis-lacto and neolacto series	Metabolism	Glycan biosynthesis and metabolism	23	0.536175	1.579291	0.018421	0.142932
KO04330	Notch signaling pathway	Environmental Information Processing	Signal transduction	92	0.417449	1.545659	0.003375	0.141311
KO00514	Other types of O-glycan biosynthesis	Metabolism	Glycan biosynthesis and metabolism	40	0.479526	1.542353	0.015912	0.134869
KO04071	Sphingolipid signaling pathway	Environmental Information Processing	Signal transduction	180	0.383745	1.526666	0	0.122295
KO04392	Hippo signaling pathway-multiple species	Environmental Information Processing	Signal transduction	36	0.476093	1.512145	0.019802	0.119366
KO00531	Glycosaminoglycan degradation	Metabolism	Glycan biosynthesis and metabolism	23	0.521457	1.486359	0.037662	0.129264
KO04010	MAPK signaling pathway	Environmental Information Processing	Signal transduction	386	0.351289	1.444061	0.001005	0.139215
KO04630	Jak-STAT signaling pathway	Environmental Information Processing	Signal transduction	163	0.36501	1.442168	0.009395	0.130223

Note: ES: enrichment score; NES: normalized enrichment score; NOM p-val: nominal *p*-value; FDR q-val: false discovery rate.

## Data Availability

The data that support the findings of this study are available from the corresponding author, Xiaohui Dong, upon reasonable request.

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
