# Peer review of "Transcriptomic Analysis of Liver in Silver sillago, Sillago sihama Fed with High-Level Low-Gossypol Cottonseed Meal in Replacement of Fishmeal Diet"

_animals, 2023, doi:10.3390/ani13071194_

Round 1

Reviewer 1 Report

This manuscript discusses the effect of replacing fish meal with high-level low-gossypol cottonseed meal on the growth and liver transcriptome of Sillago sihama. The results are well documented but some problems remain:

1. The language needs to be double-checked for singular and plural usage, as well as some grammar-related issues.

2. Line82 uses % and line91 uses percent. It is recommended to use the same way.

3. This manuscript does not provide the methods used to anesthetize and kill the fishes.

4. Statistical analysis: Have you tested the data for normality? SD varies widely between groups in some results.

5. The manuscript does not mention the method for measuring gossypol.

6. The results are rather numerous and a graphical summary is recommended.

7. Why seawater salinity should be adjusted to 6-8

8. Why not provide formulas for WGR, SGR and SR, etc.

Author Response

请参阅附件。

Reviewer 2 Report

Dear Editor,

The manuscript entitled “Transcriptomic analysis of liver in silver sillago, Sillago sihama fed with high-level low-gossypol cottonseed meal in replacement of fishmeal diet” by Hao Liu et al. presents a study focused to assessment of growth performance, quality and liver function of juvenile Sillago sihama through growth indices, whole-body composition, liver histology, and RNA-seq, after they were fed a formulated diet with 64% low-gossypol cottonseed meal (LCSM) for 56 days, compared to those fed a traditional FM-based diet. The authors conclude that feeding high levels of LCSM in S. sihama are harmful for the growth of the organism and can harm the liver's structural integrity as well as obstruct normal metabolism of amino acids, lipids, and carbohydrates. Therefore, it is not recommended to substitute LCSM for high levels of FM in the diet of S. sihama

Τhe manuscripts’ objects are interesting, it well written in a comprehensive way and the findings are interesting and justified.  Therefore, the manuscript could be accepted for publication after some minor revisions:

1.      In table 2, the proximate composition part should be somehow highlighted.

2.      Pages 4-5, paragraphs 2.4 and 2.6: Please describe how RNA was isolated for transcriptome analysis and describe how you performed sample quality checking methods (e.g. Bioanalyzer?).

3.      Figure 1: initial and final weigh diagrams can be merged to one subfigure. Please number subfigures (a,b,c,…) and their titles in the legend. The figure is not easily readable.

4.      Figure 2: Please number subfigures (a,b,c,…) and their titles in the legend. The figure is not easily readable.

5.      Figure 4: Please add a more descriptive legend.

6.      Figure 6: Please replace the yellow color. Is not readable.

7.      The authors mention table S2, however the manuscript did not have supplementary material and there is no reference of table S1 in the text.

8.      The conclusion section need more information and a clear take home message for the readers.

9.      Please check the references again and remove references in Chinese (e.g. Cao J. et al, 2008; Huang Y et al., 2015; Lin X et al, 2020; etc) since I cannot read it (and a great portion of Animals readers I suppose). Please provide alternative references in English. Also there are references with doi which could not be found (e.g. Huang Y et al., 2013; Liu J et al., 2010a (which is also the same as Liu J et al., 2010b); etc. Also, Son J et al., 2016 seems incomplete.
